# Effects of Lactoferrin and *Lactobacillus* Supplementation on Immune Function, Oxidative Stress, and Gut Microbiota in Kittens

**DOI:** 10.3390/ani14131949

**Published:** 2024-07-01

**Authors:** Hao Dong, Weiwei Wang, Qianqian Chen, Xiaohan Chang, Longjiao Wang, Shuxing Chen, Lishui Chen, Ran Wang, Shaoyang Ge, Wei Xiong

**Affiliations:** 1Food Laboratory of Zhongyuan, Luohe 462300, China; donghao980623@163.com (H.D.); wangweiwei@zyfoodlab.com (W.W.); 2020920269@stu.haut.edu.cn (Q.C.); changxiaohan@zyfoodlab.com (X.C.); chengshuxing1@163.com (S.C.); chlishui@sina.com (L.C.); 2Henan Zhiyuan Henuo Technology Co., Ltd., Luohe 462300, China; longjiaowang2019@163.com; 3Key Laboratory of Precision Nutrition and Food Quality, Department of Nutrition and Health, China Agricultural University, Beijing 100083, China; wangran@cau.edu.cn (R.W.); geshaoyang@foxmail.com (S.G.)

**Keywords:** kittens, immune deficiency, lactoferrin, *Lactobacillus*, gut microbiota

## Abstract

**Simple Summary:**

Kittens often suffer from weak immune systems, making them more prone to infections and diseases, which can severely affect their health and growth. This study explored whether adding certain dietary supplements, specifically lactoferrin and a type of probiotic called *Lactobacillus plantarum*, could help improve the health of kittens. The results showed that these supplements significantly boosted the kittens’ immune systems, making them better at fighting off infections. We also found that the supplements helped reduce stress in the body and improved the health of the kittens’ intestines by increasing the number of beneficial bacteria. Additionally, the kittens showed noticeable improvements in their physical condition, including better coat health and overall vitality. These findings suggest that using lactoferrin and *Lactobacillus plantarum* as dietary supplements can be an effective way to improve the health of kittens, reducing the need for antibiotics and their associated risks. This research provides valuable insights for pet owners and veterinarians on how to better care for kittens, ensuring they grow up healthy and strong.

**Abstract:**

Immune deficiency is a prevalent issue among kittens, severely threatening their health and development by increasing susceptibility to infections and diseases. This study investigates the effects of dietary supplements containing lactoferrin and *Lactobacillus plantarum* (*L. plantarum*) on the immune function, intestinal health, and microbiota composition of kittens. The results demonstrate that these supplements significantly enhance immune responses, with immunoglobulin A (IgA) levels increasing by 14.9% and IgG levels by 14.2%. Additionally, there was a notable 28.7% increase in catalase activity, indicating a reduction in oxidative stress. Gastrointestinal (GI) health improved markedly, evidenced by increased populations of beneficial bacteria such as *Lactobacillus*, which rose from 4.13% to 79.03% over the study period. The DNC group also showed significant reductions in pro-inflammatory cytokines, including decreases of 13.94% in IL-2, 26.46% in TNF-α, and 19.45% in IFN-γ levels. Furthermore, improvements in physical conditions were observed, including enhanced coat condition and mental status. These findings underline the potential of lactoferrin and *L. plantarum* as effective dietary interventions to improve kitten health, thereby reducing dependency on antibiotics and mitigating associated risks. This research provides a scientific foundation for optimizing nutritional management practices to enhance the overall vitality of kittens during their critical growth phases.

## 1. Introduction

With the rise in global living standards and evolving consumer habits, pets, particularly cats, have become beloved family members, providing companionship and joy [1]. This trend is reflected in the significant growth of the global pet product market, which reached USD 184.4 billion in 2023 [2]. However, despite the increasing population of kittens, they face a major challenge: immune deficiency. This heightens kittens’ susceptibility to infections and diseases and is a prevalent issue that significantly impacts their health and survival rates [3,4].

In the initial months post-birth, kittens’ immune systems are immature, relying primarily on maternal antibodies obtained through colostrum. Weaning negatively impacts the immune system, leading to lower levels of protective antibodies and higher susceptibility to infections [5]. Immunodeficient kittens often experience slow weight gain, poor coat condition, lethargy, and gastrointestinal issues such as diarrhea and vomiting [6,7,8,9]. These symptoms result from their fragile intestinal barriers and inability to effectively clear enteric pathogens. Although antibiotics are commonly used to prevent and treat infections in kittens with weak immune systems, overuse can lead to antibiotic resistance and disrupt the gut microbiota, which is essential for a healthy immune system [10]. Addressing nutritional deficiencies and managing chronic health issues are also crucial for maintaining overall vitality and health in immunocompromised kittens [11]. In light of these challenges, alternative approaches such as lactoferrin and probiotics have shown promise in enhancing immune function and mitigating the reliance on antibiotics.

Lactoferrin, a natural protein involved in iron metabolism and host defense, supports immune function and skin health through various mechanisms, including iron-binding, immunomodulation, and antimicrobial activity [12]. The results of clinical trials in humans have demonstrated that lactoferrin inhibits the growth and proliferation of infectious pathogens and provides a defense against microbial invasion through mucosal barriers. These findings suggest that lactoferrin, by participating in specific immune response mechanisms, has the potential to enhance the immunological resilience of companion animals [13]. Additionally, lactoferrin has been shown to suppress pro-inflammatory cytokines, including IL-1, IL-6, and TNF-α in vertebrates, thereby controlling inflammation and preventing immune system overactivation that can cause tissue damage. By binding to lipopolysaccharides (LPSs) on bacterial surfaces, lactoferrin inhibits LPS-induced immune cell activation, reducing pro-inflammatory cytokine expression [14]. Lactoferrin also positively influences the human gut microbiota by promoting beneficial bacteria and inhibiting pathogenic microbes [15]. In vivo experiments on mice have shown that lactoferrin, abundant in Milk-exosomes, stimulates dermal papilla cell proliferation and hair growth through the Erk/Akt and Wnt signaling pathways. This highlights lactoferrin’s potential in preventing hair loss and promoting hair growth. This suggests that lactoferrin can effectively promote hair growth in kittens by enhancing dermal papilla cell proliferation and activating critical signaling pathways involved in hair follicle development [16].

Probiotics such as lactobacilli play a crucial role in enhancing gut health and boosting immunity, given the vital importance of the gut microbiota for overall health. Strains such as *Lactobacillus*(*L.*) *rhamnosus*, *L. plantarum*, and *L. reuteri* have been shown to significantly improve gut performance. Probiotics such as *L. rhamnosus*, *L. plantarum*, and *L. reuteri* have been shown to significantly improve gut performance in humans and various animal species. However, it is crucial to consider the unique dietary and physiological characteristics of cats, which are strict carnivores. In cats, studies have demonstrated that probiotics can modulate the gut microbiota and improve gastrointestinal health. For instance, Lubbs et al. reported that *L. acidophilus* supplementation altered fecal microbiota composition and increased the abundance of beneficial bacteria in cats [17]. Additionally, Bybee et al. observed that supplementation with the probiotic *Enterococcus faecium* SF68 improved fecal consistency and significantly reduced the duration of diarrhea in cats suffering from this condition [18]. These findings suggest that probiotics can play a beneficial role in feline gut health, albeit through mechanisms that may differ from those observed in omnivores. Further research demonstrates that *Lactobacillus* strains boost protein levels, enhancing immune function. Multi-strain *Lactobacillus* products increase immune responses in lab animals, and these strains also improve gastrointestinal comfort and modulate blood parameters and inflammatory markers [19,20]. Additionally, lactobacilli improve skin quality through the gut–skin axis. For example, *L. rhamnosus* SP1 improves dermal gene expression related to insulin signaling and reduces acne in adults [21]. Fermented soymilk containing *L. casei* has been found to improve both skin condition and the gut microbiota [22]. Moreover, *L. plantarum*, *L. reuteri*, and *L. rhamnosus* reduce clinical signs of atopic dermatitis and inflammatory markers in adults [23]. *L. rhamnosus* Probio-M9 has been shown to enhance psychological and physiological quality of life by positively affecting gut microbiota and metabolites in healthy postgraduate students (in master’s or doctoral programs) over 20 years old [24]. These findings collectively underscore the potential of probiotics such as lactobacilli to significantly enhance health and immunity through their multifaceted benefits for gut and skin health.

Lactoferrin and probiotics have significant potential as dietary supplements to enhance immunity in kittens, improving their resistance to infections and overall health by supporting gut health and modulating the immune response [25].

The aim of this study is to investigate the impact of dietary supplements containing lactoferrin and *L. plantarum* CR12 on the immune function, intestinal health, and microbiota composition in pet cats. We explore the potential of scientific nutritional management to enhance health during the critical growth phases of kittens. The significance of this research lies not only in its implications for improving kittens’ quality of life but also in reducing the concerns of pet owners. Additionally, we examine the positive effects of these functional substances on kitten health and highlight the necessity of employing a scientific approach in the routine nutritional management of pets. This approach ensures the careful selection and application of these substances, ultimately supporting optimal health outcomes.

## 2. Materials and Methods

### 2.1. Animal Ethics Statement

All experimental procedures were authorized by the Animal Care and Use Committee prior to animal experimentation (Aw11104202-5-3) and were performed following the guidelines of the Institute of Nutrition and Health, China Agricultural University.

### 2.2. Experimental Design

Twelve cats aged 2–3 months were selected, comprising six males and six females. The study lasted 35 days, including a 7-day acclimatization period and a 28-day trial period. Kittens were divided equally into two treatment groups (*n* = 6/group) based on initial body weight, age, and sex: control (CON; dietary nutritional combinations not added) and experimental (DNC; dietary nutritional combinations); DNC included 500 mg/kg lactoferrin and 1 × 10^10^ CFU/kg *L. plantarum* CR12.

The test animals were not vaccinated or dewormed prior to the study, and the kittens did not receive any antibiotics or other drugs that could have caused changes in their intestinal microbiota during the 28 days prior to the study. The kittens were housed individually in cages measuring 80 × 72 × 65 cm. The temperature was controlled at 25 °C with a humidity range of 40–60%. They were allowed to socialize and exercise outside the cage for approximately 1 h per day. A variety of toys and cat teasers were provided for human interaction. Fresh food was available for *ad libitum* feeding at 9:00 a.m. and 6:00 p.m. every day, and fresh water was freely available. The food and water bowls were cleaned daily before feeding, and the cages and feeding areas were sterilized weekly. We recorded the daily food intake (in grams per day) of each kitten. The daily feed quantity for kittens should be around 50 g.

### 2.3. Diet

Detailed information on the ingredients and chemical composition of the expanded dry basal diets consumed by the kittens is shown in Table 1. The expanded dry basal diets were manufactured and formulated to meet the Association of American Feed Control Officials (AAFCO) nutritional recommendations for kittens [26]. The ingredient composition of both experimental diets was the same except for the addition of DNC.

### 2.4. Sample Collection

#### 2.4.1. Blood Collection and Analyses

The kittens underwent an overnight fasting regimen, and on day 28 of the study, 2 mL of blood was obtained from a forelimb vein of each cat. A portion of the blood specimen was collected into centrifuge tubes containing ethylenediaminetetraacetic acid (EDTA) as an anticoagulant for the determination of routine blood indices. The remainder of the specimen was placed in anticoagulant-free serum separator tubes and allowed to coagulate at 28 °C for 30 min, and thereafter subjected to centrifugation at 3000× *g* for 10 min at 4 °C. After centrifugation, an equivalent volume of supernatant was aliquoted into microcentrifuge tubes and stored at −80 °C until analysis. The cats’ serum biochemical indices were determined using a blood biochemistry analyzer (BS-240 Vet, Mindray Animal Medical Tech., Shenzhen, China). The serum antioxidant capacity was determined using a kit (Nanjing Jiancheng Bioengineering Research Institute, Nanjing, China) according to the manufacturer’s instructions to assess superoxide dismutase (SOD), total antioxidant capacity (T-AOC), malondialdehyde (MDA), glutathione peroxidase (GSH-Px), and catalase (CAT) levels. Immunoglobulin concentrations (IgA, IgG, IgM) were quantified using a turbidimetric kit (Nanjing Jiancheng Bioengineering Research Institute, Nanjing, China) according to the provided protocol. Tumor necrosis factor-alpha (TNF-α), interferon-gamma (IFN-γ), interleukin-2 (IL-2), and interleukin-4 (IL-4) levels were measured using ELISA kits (Shanghai Enzyme Link Bio-technology Co., Ltd., Shanghai, China) according to the manufacturer’s instructions. Serum parameters indicating intestinal barrier function were determined using the WST chromogenic D-lactic acid (D-LA) level assay kit and the colorimetric diamine oxidase (DAO) level assay test kit (Shanghai Enzyme Link Bio-technology Co., Ltd., Shanghai, China), respectively, in accordance with the manufacturer’s instructions, to assess serum levels of D-LA and DAO.

#### 2.4.2. Hair Collection and Analyses

##### Cat Hair Collection

The collection of cat hair for experimental analysis was conducted under a standardized protocol. Prior to collection, kittens were acclimatized to the environment to reduce stress-induced variability in hair structure. Hair samples were collected using a non-invasive brushing technique, employing a sterile fine-toothed comb to gently retrieve hairs from the dorsal and lateral aspects of the torso, areas typically abundant in representative hair samples. The collected hairs were then placed on a dated specimen container. Each container was stored at −20 °C to preserve the structural integrity of the hair until analysis.

##### Kitten Hair Scoring Basis and Method

The scoring of cat hair was conducted by professionals possessing expertise in the dermatological assessment of felid fur and skin conditions, utilizing a holistic evaluation method that integrates both visual and tactile assessments. The assessment covered hair samples from the facial area, limbs, coat, and tail of the felines, referencing the scoring guidelines established by the Fédération Internationale Féline (FIFe), an international federation of cat organizations [27]. These guidelines were developed to facilitate a quantitative evaluation of the characteristics of kitten fur. The scoring process was conducted in a blinded manner to ensure objectivity and minimize bias.

##### Hair Sample Preparation

Scanning electron microscope (SEM) analysis involved a preparatory process for hair samples, including dehydration, drying, and sputter-coating with gold. The samples were examined at magnifications ranging from 500× to 10,000×, enabling detailed visualization of the hair cuticle structure, shaft morphology, and surface irregularities. Digital images were captured and analyzed for hair scale thickness, hair scale length, and hair diameter of the kittens’ in the captured electron microscope images using Fiji-ImageJ software (ImageJ, version 1.52a, Madison, WI, USA) to quantify structural features [28]. The application of professional scoring and advanced microscopy technology allowed for a comprehensive microscopic analysis of coat health, providing a detailed assessment of the kitten’s coat quality.

### 2.5. Analysis of 16S rRNA Gene Diversity of Microbiota in the Intestinal Tract of Kittens

Prior to the commencement of the feeding regimen at day 0 of the experiment, kittens were allowed to defecate naturally. Following defecation, approximately 1–2 g of sample was aseptically collected from the fresh feces of each kitten using sterile swabs. The collected fresh fecal samples were then placed into pre-prepared sterile cryovials. The same collection procedure was repeated at the conclusion of the experiment. Subsequently, the sampled cryovials were immediately transferred to a −80 °C freezer for storage, pending further DNA extraction and analysis.

Total DNA extraction from the microbial community was conducted following the protocols provided by the E.Z.N.A.^®^ soil DNA kit (Omega Bio-tek, Norcross, GA, USA). The quality of extracted DNA was assessed using 1% agarose gel electrophoresis, while DNA concentration and purity were determined using a NanoDrop™ 2000 spectrophotometer (Thermo Scientific, Waltham, MA, USA), where the A260/280 ratio of DNA should be greater than 1.8 and the A260/A230 ratio greater than 2.0. The V3-V4 regions of the 16S rRNA gene were amplified via PCR using primers 338F (5’-ACTCCTACGGGAGGCAGCAG-3’) and 806R (5’-GGACTACHVGGGTWTCTAAT-3’). PCR products from the same sample were pooled, purified, and quantified using a Quantus™ Fluorometer (Promega, Madison, WI, USA). Library construction was performed with the NEXTflex™ Rapid DNA-Seq Kit (Bio Scientific, San Diego, CA, USA). The library was then sequenced on an Illumina NovaSeq platform, generating 250 bp paired-end reads. Subsequent read filtering was conducted using QIIME quality filters (QIIME 1.9.1) as described by Caporaso et al. [29]. The sequence data had been submitted to the SRA database under accession number SUB14566918.

### 2.6. Kitten Feces Scoring Basis and Method

The Cat Feces Scoring Method standardizes the collection and assessment of fecal samples, ensuring accurate health evaluations through consistent practices. Initially, felines are acclimated within the collection environment to mitigate stress-related alterations in fecal consistency. A standardized diet is implemented for 48 h to maintain uniform fecal composition. Collection requires the use of aseptic gloves and instruments. The scoring system assesses consistency, as outlined in the modified Bristol Stool Scale [30]. This rigorous approach significantly improves the precision of fecal diagnostics, enhancing veterinary health surveillance.

### 2.7. Methodology for Evaluating Kitten Vitality

According to the Kitten Mental Status Scoring Guidelines in Table 2, the kittens were evaluated and scored. For the assessment of mental state, the protocol involved bi-weekly evaluations conducted by professionals. These evaluations were designed to systematically observe and score the kittens’ behavioral and psychological responses, thereby ensuring a comprehensive understanding of their mental well-being throughout the study period.

### 2.8. Statistical Analyses

The experimental findings were analyzed using IBM Statistical Package for Social Sciences (SPSS) software (version 22.0; SPSS Inc., Chicago, IL, USA). Student–Newman–Keuls multiple range tests were implemented to evaluate variation among treatments. Significance was established at *p* ≤ 0.05; the data are expressed as the mean ± standard deviation (SD). The graphics in this study were produced using GraphPad (GraphPad Prism., version 8.0.2, Boston, MA, USA). A Spearman correlation heatmap based on the Spearman correlation coefficients among the bacterial profiles and serum indices was produced using R software (R Core Team, version 2.15.3, Vienna, Austria).

## 3. Results

### 3.1. Physical Condition Assessment

Several indicators reflecting the physical condition of cats, including coat condition, mental status, fecal scores, and body weight, are shown in Figure 1, with both the CON and DNC groups showing similar baseline values initially. Coat condition and mental status indices improved significantly in the DNC group at 14 and 28 days compared to the CON group. Specifically, the DNC group showed significant improvement in coat condition at 14 days (*p* = 0.04) and 28 days (*p* = 0.01) (Figure 1A). Similarly, enhancements in mental status were significant at 14 days (*p* = 0.02) and 28 days (*p* = 0.01) (Figure 1B). In terms of fecal scoring, the DNC group demonstrated significantly better scores at 28 days (*p* = 0.002) (Figure 1C). The kittens had an average body weight of 839 ± 75.3 g at the beginning of the study. There was no significant difference in the mean body weight of the kittens in each group at the start of the intervention. Weight measurements showed a steady increase in both groups throughout the study, with no significant differences observed at any point, but the DNC group showed a trend of faster weight gain (*p* > 0.05) (Figure 1D).

### 3.2. Hematological Indicators

#### 3.2.1. Complete Blood Count

Table 3 presents the effects of DNC on cats’ CBC. The DNC group exhibited a lower total white blood cell count (16.33 ± 4.10 × 10^9^/L) compared to the CON group (21.50 ± 3.81 × 10^9^/L), with the difference being statistically significant (*p* < 0.05). Other hematological parameters did not show significant differences between the groups (*p* > 0.05). In addition, the total white blood cell in the DNC group was within the normal range, whereas in the CON group it exceeded the normal range.

#### 3.2.2. Serum Biochemical Indices

The serum biochemical parameters of the CON and DNC groups are presented in Table 4. Significant differences were observed in the levels of aspartate aminotransferase (AST) and creatine kinase (CK) between the two groups. The DNC group exhibited significantly lower AST levels (12.51 ± 5.20 U/L) compared to the CON group (22.14 ± 3.86 U/L), with the difference being highly significant (*p* < 0.01), although the values for both groups remained within the normal reference range. Additionally, the DNC group showed significantly lower CK levels (373.31 ± 65.22 U/L) compared to the CON group (502.42 ± 95.31 U/L), with the difference reaching statistical significance (*p* < 0.05). In the CON group, CK levels were significantly higher than the normal reference range. There were no significant changes in other serum biochemical parameters.

#### 3.2.3. Immunoglobulin Parameters

The immunoglobulin levels (IgA, IgM, and IgG) of the CON and DNC groups are presented in Figure 2. The IgA levels in the DNC group (1.298 g/L) were 14.9% higher than those in the CON group (1.130 g/L), with the increase being statistically significant (*p* = 0.004). The IgM levels showed no significant difference between the DNC group (0.9336 g/L) and the CON group (0.772 g/L) (*p* = 0.118), indicating no significant impact from the DNC intervention. Conversely, IgG levels were elevated by 14.2% in the DNC group (6.638 g/L) compared to the CON group (5.811 g/L), with a statistically significant difference (*p* = 0.02).

#### 3.2.4. Antioxidant Parameters

Figure 3 illustrates the oxidative stress markers and antioxidant enzyme activities in the CON and DNC groups. The DNC group showed a marked reduction in MDA levels, which were 11.3% lower (4.928 nmol/mL) compared to the CON group (5.558 nmol/mL), indicating significantly reduced lipid peroxidation (*p* = 0.003). CAT activity was notably elevated by 28.7% in the DNC group, with levels reaching 34.56 U/mL, compared to 26.85 U/mL in the CON group (*p* = 0.0001). Similarly, GSH-PX activity was enhanced by 22.2% in the DNC group, showing a significant increase to 137.5 U/mL from 112.5 U/mL in the CON group (*p* = 0.012), indicating better peroxide reduction capabilities. In contrast, T-AOC and SOD activities did not differ significantly between the groups (*p* > 0.05). These findings collectively suggest that the DNC treatment effectively enhanced specific antioxidant defenses and reduced oxidative stress in the DNC group.

#### 3.2.5. Cytokine Levels

Figure 4 presents the cytokine levels in the CON and DNC groups. The DNC group showed a significant reduction in IL-2 levels, which decreased by 15.4% to 206.8 pg/mL, compared to 244.5 pg/mL in the CON group (*p* = 0.002). IFN-γ levels were also notably lower in the DNC group, with a 30.4% reduction to 36.67 pg/mL, relative to 52.68 pg/mL in the CON group (*p* = 0.0019). Additionally, TNF-α levels dropped by 18.6% in the DNC group, reaching 63.10 pg/mL, in contrast to 77.53 pg/mL in the CON group (*p* = 0.005). Conversely, IL-4 levels showed no significant variation between the two groups. These results collectively suggest that the DNC treatment effectively lowered several pro-inflammatory cytokines, indicating a potential anti-inflammatory effect.

#### 3.2.6. Gut Barrier Function Parameters

The levels of D-LA and DAO were assessed, as depicted in Figure 5. The D-LA levels showed no significant difference between the CON and DNC groups (*p* = 0.441). Conversely, DAO levels were markedly lower in the DNC group (1.990 U/mL) compared to the CON group (2.603 U/mL), with a significant decrease of 23.5% (*p* = 0.0008).

### 3.3. Hair Condition Analysis

The hair structural characteristics were evaluated using SEM, and the results are shown in Figure 6. The hair scale thickness was significantly reduced in the DNC group at day 28 (0.362 μm) compared to the CON group at day 0 (0.807 μm) and day 28 (0.666 μm), with reductions of 55.2% and 45.6%, respectively (*p* < 0.001) (Figure 6a). In the DNC group, the hair scale length at day 28 (4.477 μm) was significantly shorter than that in the CON group at day 0 (9.945 μm) and day 28 (6.570 μm), with reductions of 55.0% and 31.8%, respectively (*p* < 0.01) (Figure 6b). An increase in hair diameter was observed in the DNC group at day 28 (39.284 μm) compared to the CON group at day 0 (30.596 μm) and day 28 (32.679 μm), showing increases of 28.4% and 20.2%, respectively (*p* < 0.001) (Figure 6c).

### 3.4. Fecal Microbiota Composition

The alpha-diversity indices of the fecal microbiota are shown in Table 5. The Ace index, which estimates species richness, was significantly lower in the DNC group at day 28 (56.67 ± 10.55) compared to the CON group at day 0 (100.50 ± 35.18), CON group at day 28 (106.80 ± 8.95), and DNC group at day 0 (109.00 ± 8.10) (*p* < 0.01). In the DNC group, the Chao1 index showed a significant decrease at day 28, registering 54.27 ± 8.12, compared to the CON group at day 0 (101.60 ± 35.35), the CON group at day 28 (103.90 ± 9.63), and the DNC group at day 0 (110.8 ± 10.97) (*p* < 0.01). The Sobs index, which measures observed species, showed a notable decrease in the DNC group at day 28 (50.80 ± 8.67) when compared to the CON group at day 0 (93.60 ± 33.02), the CON group at day 28 (94.60 ± 10.60), and the DNC group at day 0 (96.20 ± 7.59) (*p* < 0.01). There were no significant differences observed among the groups in respect of the Coverage, Shannon, and Simpson indices.

The investigation on the composition and distribution of the feline fecal microbiota, utilizing 16S rRNA sequencing, is detailed in Figure 7. In the fecal samples from four distinct groups of kittens, a total of 315 OTUs were identified. Among these, 56 OTUs were found across all groups, signifying the existence of a core microbiota composition ((A) in Figure 6). PCoA highlighted distinct microbial community patterns for each group. Notably, a distinct differentiation was observed on the PCoA plot between DNC28 and other groups ((B) in Figure 6). (C) in Figure 6 illustrates the relative abundance of Firmicutes and Actinobacteriota at various time points. Firmicutes exhibited a significant increase in DNC28 (89.03%) compared to CON0 (74.1%), CON28 (78.3%), and DNC0 (60.1%). This suggests that the DNC intervention led to an elevated proportion of Firmicutes over the 28-day period. Conversely, Actinobacteriota showed a marked reduction in the DNC group at day 28 (9.59%) when compared to CON0 (23.23%), CON28 (17.1%), and DNC0 (32.13%), indicating a significant decrease in Actinobacteriota abundance following the DNC treatment. The relative abundances of various bacterial genera in the fecal microbiota across the four groups are illustrated in (D) in Figure 6. The relative abundance of *Lactobacillus* increased dramatically from 4.13% in the CON0 group to 79.03% in the DNC28 group. This substantial rise indicates significant enrichment of *Lactobacillus* due to the DNC intervention over the 28-day period. Conversely, the relative abundance of *Megasphaera* showed a marked decline over time. It decreased from 22.86% in the CON0 group to 10.09% in the CON28 group, and eventually to 0% in the DNC28 group. The relative abundance of *Bifidobacterium* showed fluctuations over the study period. In the CON0 group, it was 11.73%, which increased to 17.34% in the DNC0 group. After 28 days, it decreased to 8.16% in the CON28 group and slightly increased to 9.44% in the DNC28 group. The relative abundance of *Blautia* decreased significantly from 20.22% in the CON0 group to 7.62% in the CON28 group, and it was nearly eliminated in the DNC28 group. Similarly, *Collinsella* showed a decrease from 11.17% in the CON0 group and 14.16% in the DNC0 group to 8.75% in the CON28 group, approaching 0% in the DNC28 group. *Enterococcus* exhibited significant changes, starting at 1.97% in the CON0 group, increasing to 16.19% in the CON28 group, and then dropping to nearly 0% in the DNC28 group. These results suggest that the DNC treatment led to a substantial reduction in the abundances of *Blautia*, *Collinsella*, and *Enterococcus* over the 28-day period.

### 3.5. Correlation Analysis

Figure 8 illustrates the correlation patterns between the 20 most abundant bacterial taxa and various serum indices. *Lactobacillus* exhibits a strong positive correlation with CAT (*p* = 0.04, *r* = 0.81), and also shows positive correlation with IgA, IgG, and IgM. However, there is a significant negative correlation with DAO (*p* = 0.02, *r* = −0.72), MDA (*p* < 0.01, *r* = −0.83) and IL-2 (*p* = 0.04, *r* = −0.65). *Blautia* demonstrates a positive relationship with IFN-γ (*p* < 0.01, *r* = 0.86), TNF-α (*p* < 0.01, *r* = 0.78), IL-2 (*p* < 0.01, *r* = 0.88), and MDA (*p* < 0.01, *r* = 0.88). However, it significantly negatively correlates with SOD (*p* < 0.01, *r* = −0.82), GSH-PX (*p* = 0.03, *r* = −0.69), IgM (*p* = 0.03, *r* = −0.69), IgA (*p* = 0.02, *r* = −0.73), IgG (*p* < 0.01, *r* = −0.79), and CAT (*p* = 0.03, *r* = −0.69). The presence of *Collinsella* is positively associated with DAO (*p* < 0.01, *r* = 0.89) and negatively associated with IgA (*p* = 0.05, *r* = −0.63). DAO is strongly positively correlated with *Megasphaera* (*p* = 0.03, *r* = 0.68), *Peptoclostridium* (*p* = 0.03, *r* = 0.37) and *Bacteroides* (*p* < 0.01, *r* = 0.79).

## 4. Discussion

The significant improvements in coat condition and mental status observed in the DNC group at 14 and 28 days suggest a positive influence of dietary nutritional combinations. Lactoferrin has been shown to enhance skin health by strengthening the skin barrier, reducing inflammation, and promoting the growth of beneficial gut bacteria [31]. Regarding mental status, probiotics can influence the gut–brain axis, thereby enhancing cognitive function. Lactoferrin and probiotics have been shown to modulate neurotransmitter levels and improve cognitive function in animal models [32]. The marked improvement in fecal scores observed in the DNC group aligns with research on the gastrointestinal benefits of probiotics. For example, Marelli et al. [33] found that supplementation with *L. acidophilus* improved fecal consistency and overall gut health in dogs, supporting the findings of better fecal scores in the DNC group.

The absence of significant changes in body weight between the CON and DNC groups suggests that the dietary intervention primarily influenced other health aspects without affecting weight. This is consistent with studies where probiotics did not significantly alter body weight but improved other health markers. For instance, supplementation with *L. plantarum* in weaning pigs did not result in significant weight changes but led to improvements in nutrient digestibility and fecal parameters [34]. The significant improvements in coat condition, mental status, and fecal scores following DNC supplementation in cats underscore the potential benefits of incorporating lactoferrin and probiotics into their diet.

Although baseline values for all variables were not measured on day 0, an even distribution of 2-month-old and 3-month-old animals across the groups was ensured to minimize potential baseline differences. This limitation is acknowledged, and future studies will include comprehensive baseline measurements to enhance the robustness of the findings.

The significant reduction in WBC count observed in the DNC group suggests that probiotic supplementation may influence immune function. Similar effects have been documented in other studies. For example, Robles-Vera et al. [35] found that probiotics such as *Bifidobacterium breve* and *L. fermentum* prevented increases in WBC counts and improved endothelial function in hypertensive rats, indicating their role in modulating immune responses and reducing inflammation. Additionally, probiotic strains increased natural killer (NK) cells and modulated T-helper cells in mice, highlighting the potential of probiotics to alter immune cell populations [36]. These findings suggest that probiotics can have a regulatory effect on the immune system, which may explain the observed reduction in WBC count in the DNC group.

The reduced AST levels in the DNC group indicate improved liver function or reduced liver stress, likely due to the protective effects of probiotics and lactoferrin. For instance, dietary encapsulated probiotics significantly reduced serum cholesterol, AST, and ALT levels in broilers, indicating that probiotics can enhance liver function and mitigate oxidative stress [37]. Another study demonstrated that *L. casei* supplementation in common carp significantly reduced AST, ALT, and CK levels, indicating protective effects against muscle and liver damage [38].

The intestinal microbiota is crucial for the development and function of gut-associated lymphoid tissue (GALT), including Peyer’s patches, where B lymphocytes proliferate and differentiate into plasma cells that secrete immunoglobulins. Probiotics enhance GALT development and maturation by modulating the gut microbiota, thereby increasing immunoglobulin production [39]. Lactoferrin, present in milk, exhibits immunomodulatory properties by binding to bacterial cell walls, which promotes the growth of beneficial bacteria and enhances GALT function. Additionally, lactoferrin binds to B lymphocyte receptors, facilitating their activation and differentiation into plasma cells. Probiotics also stimulate cytokine production, which boosts B lymphocyte activity and immunoglobulin production [40]. Strains such as *Lactobacillus* and *Bifidobacterium* have been shown to upregulate immune response genes, further enhancing the immune response [41].

Immunoglobulins enhance immune function by recognizing and neutralizing pathogens, thereby boosting the body’s immune defense capabilities [42]. The significant increase in IgA levels in the DNC group suggests that the supplement effectively boosts mucosal immunity. Lactoferrin and probiotics are known to enhance IgA production, which plays a crucial role in immune defense at mucosal surfaces. Bovine lactoferrin (BLf) treatment increased IgA levels in mice exposed to chronic stress, indicating its immunomodulatory properties, while *L. paracasei* MCC1849 enhanced antigen-specific IgA production, suggesting that probiotics can significantly boost mucosal immunity [43,44]. Probiotics have been shown to increase IgG levels in both animals and humans, with studies reporting that probiotic use can significantly raise plasma IgG concentrations, thereby supporting the role of probiotics in boosting systemic immunity [45,46].

Antioxidant indicators play a crucial role in animal health by reducing oxidative stress, thereby enhancing overall health and productivity [47]. The significant reduction in MDA levels in the DNC group suggests decreased lipid peroxidation, which is a marker of oxidative stress. Lactoferrin and probiotics have been shown to reduce oxidative damage by lowering MDA levels. For instance, Kim et al. [48] found that lactic acid bacteria (LAB) isolated from canine and feline feces demonstrated prominent antioxidant properties, effectively reducing oxidative stress markers such as MDA in animal models. Lactoferrin has exhibited potent antioxidative properties across various studies. It significantly reduced lipid peroxidation levels in HepG2 cells exposed to acrylamide [49]. Additionally, vaginal lactoferrin administration significantly increased total antioxidant status and CAT activity in the amniotic fluid of pregnant women [50]. Furthermore, lactoferrin supplementation increased GSH-PX activity in neonatal piglets. These findings collectively underscore the significant antioxidative potential of lactoferrin in different contexts [51].

Inflammatory cytokines play a crucial role in immune responses by mediating the activation and recruitment of immune cells to sites of infection or injury, thereby modulating inflammation and the overall immune response in animals [52]. The significant reduction in IL-2 levels in the DNC group suggests that the dietary intervention effectively modulates immune responses. *Bifidobacterium animalis* ssp. *lactis* HY8002 and *L. plantarum* HY7717 were found to enhance immune function by increasing the secretion of cytokines, including IL-2, in immunosuppressed mice [53]. Additionally, lactoferrin significantly reduced IL-6 and TNF-α levels in mice with X-ray-induced intestinal injury, highlighting its role in reducing inflammatory cytokines [54]. In the DNC group, a notable decrease in IFN-γ levels indicates a reduced inflammatory response, while the significant reduction in TNF-α levels highlights the anti-inflammatory impact of the dietary intervention. Furthermore, liposomal lactoferrin and lactoferrin–zinc complexes have been shown to activate IFN-γ, enhancing defense mechanisms against infections and reducing inflammation [55]. Additionally, lactoferrin attenuates lipopolysaccharide-stimulated inflammatory responses by lowering TNF-α levels in intestinal epithelial cells [56]. The significant reductions in IL-2, IFN-γ, and TNF-α levels in the DNC group underscore the potential benefits of dietary nutritional combinations, including lactoferrin and probiotics, in modulating immune responses and reducing inflammation.

The intestinal barrier is crucial for maintaining animal health by preventing the translocation of harmful microorganisms and their toxins from the intestinal lumen into the bloodstream, thus playing a significant role in immune response and nutrient absorption [57].

The significant reduction in DAO levels observed in the DNC group suggests that the dietary intervention enhances intestinal barrier function. Early-life supplementation with lactoferrin and probiotics significantly decreased DAO levels and improved gut health in neonatal piglets [32]. Additionally, BLF has been shown to reverse antibiotic-induced gut dysbiosis in mice by restoring normal levels of anti-inflammatory bacteria and reducing DAO levels, underscoring its role in maintaining gut barrier integrity [58].

Healthy hair in pets is essential for their overall well-being, providing protection against environmental factors, aiding in thermoregulation, and contributing to their aesthetic appearance, which can positively affect the pet–owner relationship [59]. The significant reduction in hair scale thickness and length, along with the observed increase in hair diameter in the DNC group, suggests that the dietary intervention contributed to a healthier hair structure and thicker hair strands. BLF has been shown to promote the proliferation of dermal papilla cells and enhance hair growth in mice through the Erk/Akt and Wnt signaling pathways, indicating its potential in improving hair quality [60]. Additionally, Yu et al. [61] reported that a multi-strain probiotic formula improved hair density and reduced hair loss in subjects with hair thinning, demonstrating the potential of probiotics in enhancing hair health.

The gut microbiota is a crucial metabolic “organ” influencing host metabolism, physiology, nutrition, and immune function, thereby determining the health and performance of animals [62]. The observed reductions in the Ace, Chao1, and Sobs indices in the DNC group indicate decreased microbial richness and diversity. Lactoferrin and probiotics have been shown to modulate gut microbiota diversity and composition. For instance, a study by Bellés et al. [58] demonstrated that BLF reversed antibiotic-induced gut dysbiosis in mice by restoring normal levels of beneficial bacteria and modulating gut microbiota composition. Probiotics are known to enhance the population of *Lactobacillus*. The significant increase in the relative abundance of Firmicutes and *Lactobacillus* in the DNC group suggests that the dietary intervention promoted the growth of these beneficial bacteria. Supplementation with *L. fermentum* and *L. plantarum* increased gut microbiota diversity and functionality while mitigating *Enterobacteriaceae* in a mouse model [63]. Similarly, supplementation with *L. fermentum* and *Pediococcus acidilactici* promoted growth performance, alleviated inflammation, and increased the abundance of *Lactobacillus* in weaned pigs [64]. The marked reduction in *Actinobacteria, Blautia*, *Collinsella*, and *Enterococcus* in the DNC group indicates a shift towards a healthier gut microbiota composition. The study found significant changes in the gut microbiota after 28 days of intervention, including greater homogeneity in diversity and richness and a decrease in *Blautia*. These changes must be interpreted with caution, as *Blautia*, a core genus associated with gut health, decreased possibly due to competitive exclusion by *Lactobacillus* [65]. Future studies should include a follow-up period post-intervention to assess the persistence and long-term implications of these changes. A multi-strain probiotic formula decreased the abundance of opportunistic pathogenic bacteria such as *Clostridium* perfringens in dogs with diarrhea [66]. Similarly, *L. rhamnosus* GG and *L. plantarum* JL01 reduced the relative abundance of harmful bacteria such as *Enterococcus* in weaned piglets [67]. Additionally, Almeida et al. [68] showed that lactoferrin reduced the adhesion and biofilm formation of *Bacteroides fragilis* and *Bacteroides thetaiotaomicron*, highlighting its role in modulating gut microbiota composition. The significant changes in bacterial composition, including increased relative abundance of Firmicutes and *Lactobacillus* and reduced abundance of Actinobacteriota and other pathogenic bacteria, highlight the potential benefits of dietary nutritional combinations, including lactoferrin and probiotics, in modulating gut microbiota.

This study also showed strong correlations between specific bacterial taxa and serum indices. For instance, *Lactobacillus* exhibited a strong positive correlation with IgG and CAT, and a negative correlation with DAO and MDA. Similarly, *Blautia* showed positive relationships with pro-inflammatory cytokines (IFN-γ, TNF-α, IL-2) and MDA, but negative correlations with IgA, IgG, and CAT. These findings are consistent with studies highlighting the immunomodulatory effects of probiotics and lactoferrin. For example, Fang et al. [69] demonstrated that probiotics modulate gut microbiota and immune responses, showing a positive correlation between specific bacterial genera and serum immune markers in patients with atopic dermatitis. Additionally, lactoferrin supplementation positively influences serum antioxidant capacity and iron status, further supporting the role of lactoferrin in enhancing immune function [70]. These findings are supported by recent studies demonstrating the ability of lactoferrin and probiotics to enhance gut health by promoting beneficial bacteria and reducing harmful ones.

## 5. Conclusions

This study demonstrated that dietary supplementation with lactoferrin and *Lactobacillus* can significantly enhance immune function, reduce oxidative stress, and improve gastrointestinal health in kittens. The observed increases in IgA and IgG levels indicate a strengthened immune response, while the reduction in oxidative stress markers, such as catalase activity, highlights the antioxidant properties of these supplements. Additionally, the substantial rise in beneficial bacterial populations, including *Lactobacillus*, shows a positive impact on gut microbiota composition, which is essential for overall health and disease resistance. Incorporating lactoferrin and *Lactobacillus* into the diet of kittens can effectively address immune deficiencies, improve health outcomes, and potentially reduce reliance on antibiotics, thereby lowering the risk of antibiotic resistance. This research provides a solid foundation for future studies and applications in pet health management, emphasizing the importance of nutritional strategies in enhancing the well-being and development of kittens during their critical growth phases.

## Figures and Tables

**Figure 1 animals-14-01949-f001:**
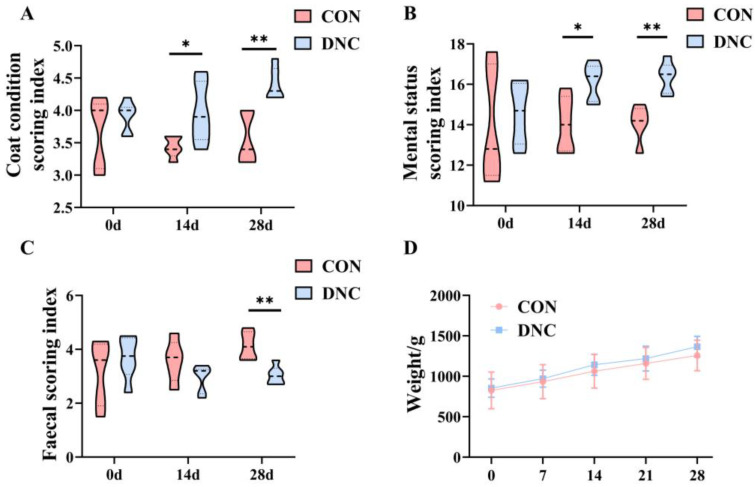
Effects of dietary nutritional combinations on physical condition in kittens. (**A**) Coat condition scoring index, (**B**) mental status scoring index, (**C**) fecal scoring index, and (**D**) weight. The significant differences between the CON and DNC groups throughout the same feeding time are expressed as * *p* ≤ 0.05 and ** *p* ≤ 0.01 according to the paired-sample *t*-test. The values are expressed as means ± SD, *n* = 6. CON, control diet, dietary nutritional combinations not added, and DNC, addition of dietary nutritional combinations.

**Figure 2 animals-14-01949-f002:**
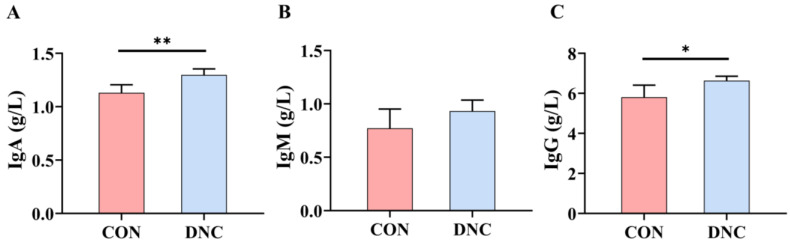
Effects of DNC on immunoglobulin parameters in kittens. (**A**) IgA, immunoglobulin A, (**B**) IgM, immunoglobulin M, and (**C**) IgG, immunoglobulin G. Significant differences between the CON and DNC groups throughout the same feeding time are expressed as * *p* ≤ 0.05 and ** *p* ≤ 0.01 according to the paired-sample *t*-test. The values are expressed as means ± SD, *n* = 6. CON, control diet, dietary nutritional combinations not added, and DNC, addition of dietary nutritional combinations.

**Figure 3 animals-14-01949-f003:**
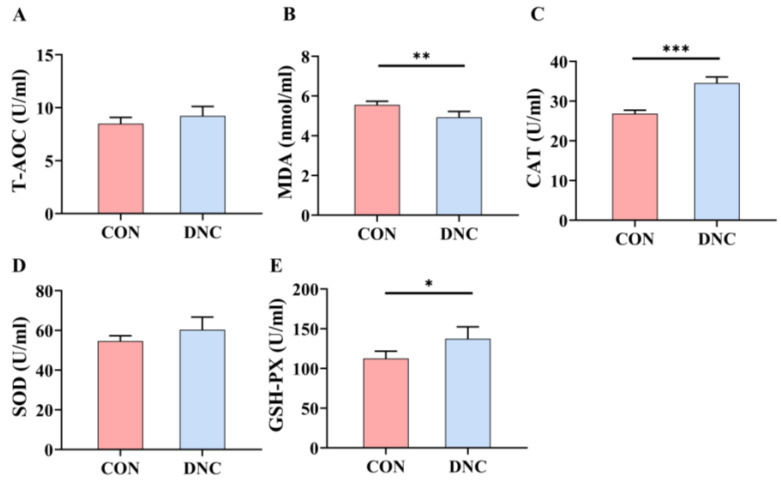
Effects of DNC on antioxidant parameters in kittens. (**A**) T-AOC, total antioxidant capacity, (**B**) MDA, malondialdehyde, (**C**) CAT, catalase, (**D**) SOD, superoxide dismutase, and (**E**) GSH-PX, glutathione peroxidase. The significant differences between the CON and DNC groups throughout the same feeding time are expressed as * *p* ≤ 0.05, ** *p* ≤ 0.01, and *** *p* ≤ 0.001 according to the paired-sample *t*-test. The values are expressed as means ± SD, *n* = 6. CON, control diet, dietary nutritional combinations not added, and DNC, addition of dietary nutritional combinations.

**Figure 4 animals-14-01949-f004:**
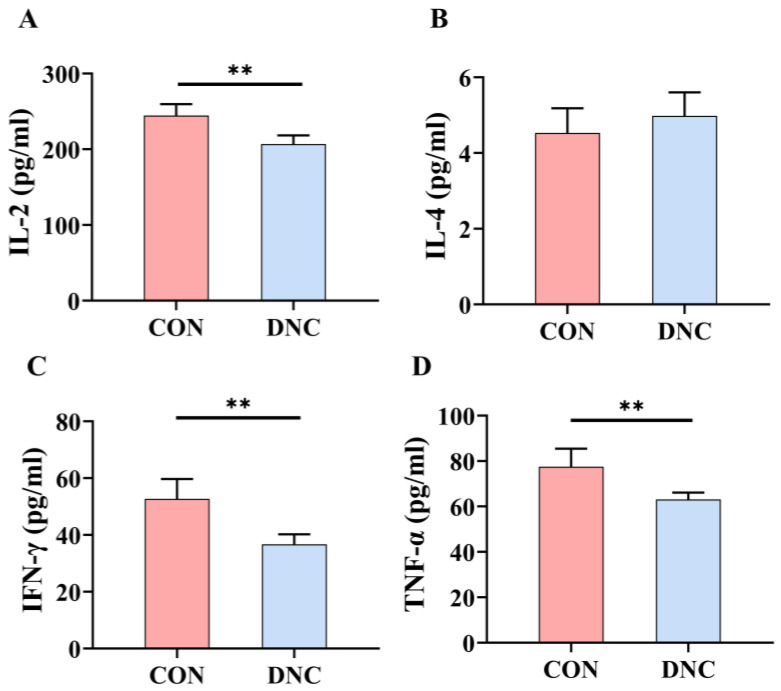
Effects of DNC on cytokine level in kittens. (**A**) IL-2, interleukin-2, (**B**) IL-4, interleukin-4, (**C**) IFN-γ, interferon-γ, and (**D**) TNF-α, tumor necrosis factor-α. The significant differences between the CON and DNC groups throughout the same feeding time are expressed as ** *p* ≤ 0.01 according to the paired-sample *t*-test. The values are expressed as means ± SD, *n* = 6. CON, control diet, dietary nutritional combinations not added, and DNC, addition of dietary nutritional combinations.

**Figure 5 animals-14-01949-f005:**
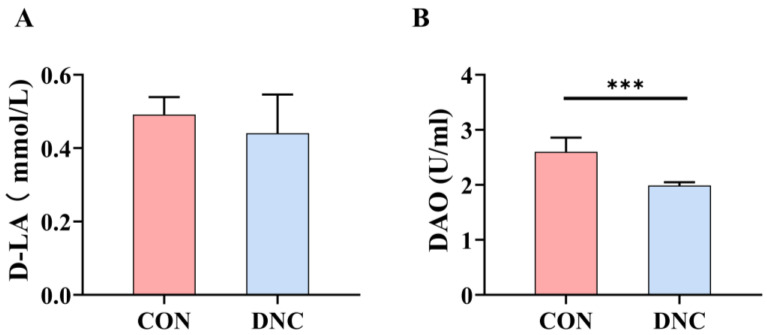
Effects of DNC on plasma intestinal barrier function parameters in kittens. (**A**) D-LA, D-lactate and (**B**) DAO, diamine oxidase. The significant differences between the CON and DNC groups throughout the same feeding time are expressed as *** *p* ≤ 0.001 according to the paired-sample *t*-test. The values are expressed as means ± SD, *n* = 6. CON, control diet, dietary nutritional combinations not added, and DNC, addition of dietary nutritional combinations.

**Figure 6 animals-14-01949-f006:**
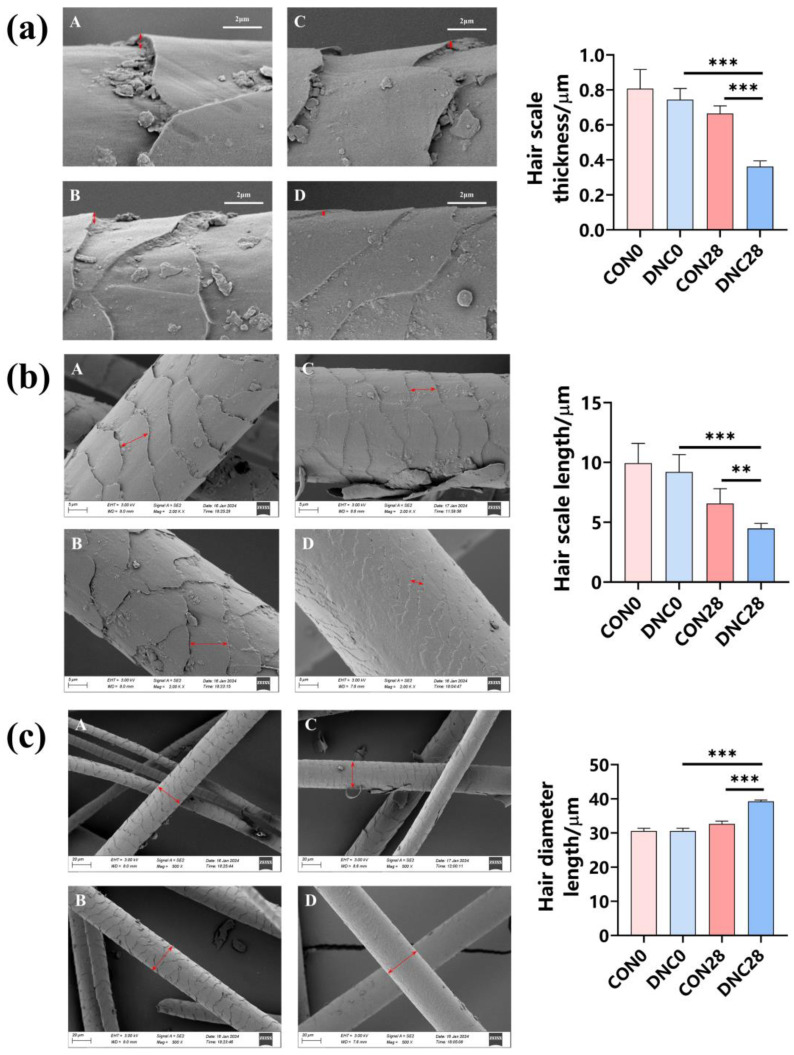
Effects of DNC on hair structural characteristics in kittens. (**a**) Hair scale thickness, (**b**) hair scale length, and (**c**) hair diameter. (**A**) CON0, (**B**) DNC0, (**C**) CON28, and (**D**) DNC28. The red arrow indicates the measuring position. The significant differences among all groups are expressed as ** *p* ≤ 0.01 and *** *p* ≤ 0.001 according to the paired-sample *t*-test. The values are expressed as means ± SD, *n* = 6. CON0, control diet, dietary nutritional combinations not added at day 0; DNC0, addition of dietary nutritional combinations at day 0; CON28, control diet, dietary nutritional combinations not added at day 28; and DNC28, addition of dietary nutritional combinations at day 28.

**Figure 7 animals-14-01949-f007:**
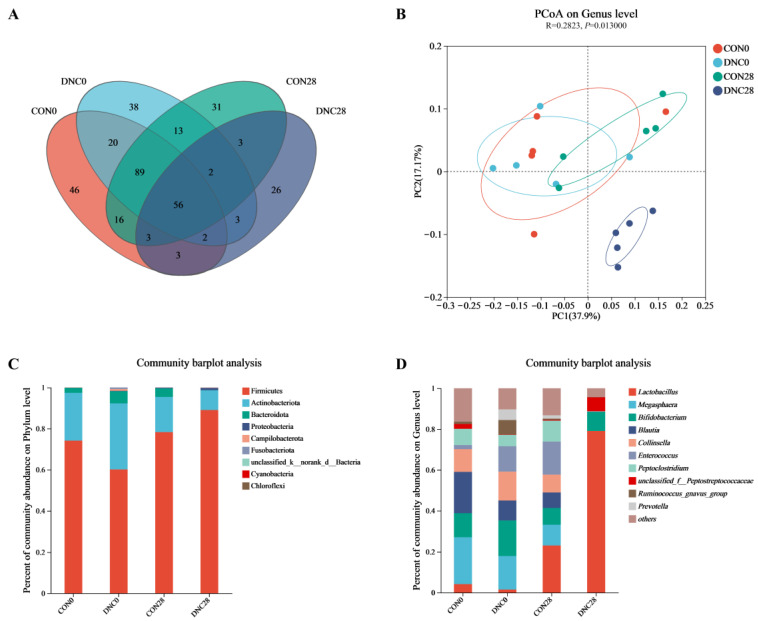
Effects of DNC on fecal microbiota composition in kittens. (**A**) Venn diagram, (**B**) principal coordinate analysis, (**C**) phylum level of bacteria, and (**D**) genus level of bacteria. The values are expressed as means ± SD, *n* = 6. CON0, control diet, dietary nutritional combinations not added at day 0; DNC0, addition of dietary nutritional combinations at day 0; CON28, control diet, dietary nutritional combinations not added at day 28; and DNC28, addition of dietary nutritional combinations at day 28.

**Figure 8 animals-14-01949-f008:**
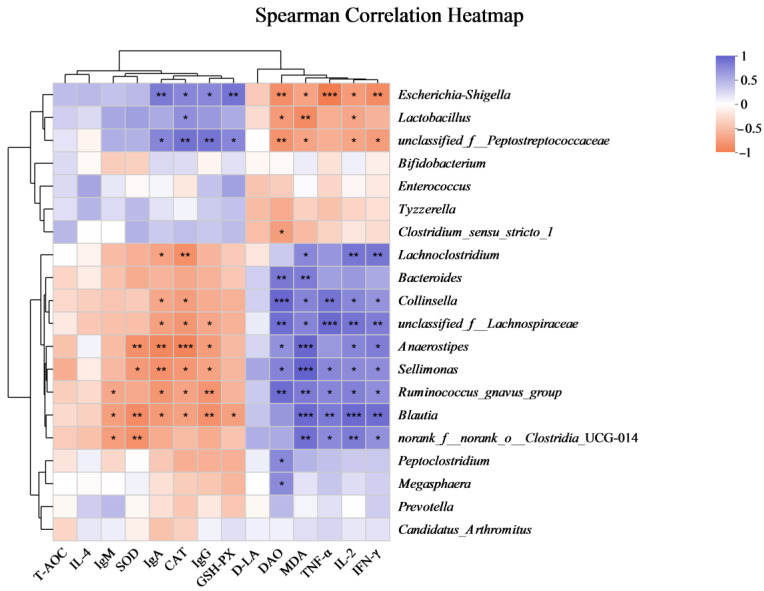
Pearson’s correlation heatmap depicting the associations between bacterial profiles and serum indices. Correlations are color-coded, with blue representing positive associations and orange representing negative ones. The significance levels are * *p* ≤ 0.05, ** *p* ≤ 0.01, and *** *p* ≤ 0.001. Data are presented as mean ± SD, with a sample size of *n* = 6.

**Table 1 animals-14-01949-t001:** Composition and nutrient content of experimental diets.

Items	Content (%)
Ingredients	
Fresh chicken	28.00
Chicken meat meal	25.00
Corn	12.00
Brown rice	10.00
Wholemeal	8.00
Chicken oil	7.40
Beetroot meal	3.00
Alfalfa pellets	2.00
Vitamins and minerals premix	1.70
Butter	1.20
Whole egg powder	1.00
Soya lecithin	0.30
Spirulina powder	0.10
Plantago	0.10
Yucca extract	0.10
Beer yeast	0.10
Nutrient content	
DM	89.74
CP	31.54
EE	19.81
Ash	9.16
WSC	0.65

^1^ Provided per kilogram diet: vitamin A, 10,000.00 IU; vitamin D3, 700.00 IU; vitamin E, 540.00 mg; vitamin K3, 0.10 mg; vitamin B1, 15.00 mg; vitamin B2, 5.00 mg; vitamin B6, 6.00 mg; vitamin B12, 0.10 mg; nicotinic acid, 90.00 mg; calcium pantothenate, 8.50 mg; D-biotin, 0.12 mg; folic acid, 0.90 mg; Fe, 80.00 mg; Cu, 15.00 mg; Mn, 7.80 mg; Zn, 80.00 mg. ^2^ DM, dry matter; CP, crude protein; EE ether extract; WSC, water soluble carbohydrates.

**Table 2 animals-14-01949-t002:** Kitten mental status scoring guidelines.

Items	2 Points	1 Point	0 Points
Eyes	Bright, clear, no discharge	Dull, cloudy, little discharge	Closed, swollen, heavy discharge
Ears	Clean, no odor, no mites or otitis	Dirty, slight odor, some mites or mild otitis	Very dirty, strong odor, many mites or severe otitis
Nose	Moist, no discharge, no congestion or sneezing	Dry, little discharge, slight congestion or occasional sneezing	Very dry, heavy discharge, severe congestion or frequent sneezing
Oral Cavity	Clean, no odor, no gum swelling or oral ulcers	Unclean, slight odor, some gum swelling or mild oral ulcers	Very unclean, strong odor, severe gum swelling or severe oral ulcers
Fur	Smooth, no matting, no fleas or skin issues	Dry, some matting, some fleas or minor skin issues	Very dry, heavy matting, many fleas or severe skin issues
Body Shape	Well-proportioned, no obesity or emaciation	Unbalanced, slight obesity or slight emaciation	Very unbalanced, severe obesity or severe emaciation
Activity	Frequent, curious, playful	Infrequent, lethargic, little play	Almost no activity, drowsy or comatose
Appetite	Keen to grab food, moderate quantity, not fussy or fed up	Occasionally poor appetite, or eats too much or too little	Long-term refusal of food or vomiting
Social	Likes to be close to people or other animals, shows friendliness and trust	Occasionally lazy or overly active	Long-term fear or aggression

^1^ Score each item for each kitten and summaries the all-item scores.

**Table 3 animals-14-01949-t003:** Effect of adding DNC on the blood routine of kittens.

Items	CON	DNC	Normal Reference Range	*p*-Value
Total White Blood Cells (10^9^/L)	21.50 ± 3.81 ^a^	16.33 ± 4.10 ^b^	5.5–19.5	<0.05
Lymphocyte Ratio (%)	56.50 ± 7.12	56.02 ± 8.68	20.0–56.0	0.92
Intermediate Cell Ratio (%)	12.02 ± 4.41	16.77 ± 7.76	2.0–19.0	0.26
Granulocyte Ratio (%)	31.48 ± 8.22	27.27 ± 8.90	25.0–85.0	0.44
Lymphocytes (10^9^/L)	12.28 ± 6.36	9.00 ± 3.37	0.8–9.0	0.30
Intermediate Cells (10^9^/L)	2.44 ± 1.08	2.73 ± 1.63	0.1–2.8	0.74
Granulocytes (10^9^/L)	6.78 ± 3.95	4.60 ± 2.73	2.1–15.0	0.31
Total Red Blood Cell s (10^12^/L)	6.44 ± 0.75	7.10 ± 1.30	4.60–10.00	0.35
Hemoglobin (g/L)	104.60 ± 10.78	106.33 ± 19.15	93–153	0.86
Hematocrit (%)	29.46 ± 2.03	31.35 ± 3.43	28.0–49.0	0.31
Mean Corpuscular Volume (fL)	46.26 ± 5.71	44.70 ± 3.76	39.0–52.0	0.60
Hemoglobin Content (pg)	16.48 ± 3.29	14.92 ± 0.61	13.0–21.0	0.28
Hemoglobin Concentration (g/L)	355.00 ± 33.75	336.67 ± 22.71	300–380	0.31
Red Cell Distribution Width SD (fL)	38.66 ± 5.65	37.80 ± 3.84	47.0–62.7	0.77
Red Cell Distribution Width CV (%)	18.04 ± 1.33	18.32 ± 1.18	14.0–18.0	0.92
Total Platelet Count (10^9^/L)	320.20 ± 165.94	497.17 ± 391.27	100–514	0.37
Mean Platelet Volume (fL)	11.08 ± 1.75	12.08 ± 2.84	5.0–12.8	0.51
Platelet Distribution Width (%)	11.20 ± 2.50	11.92 ± 3.49	0.1–30.0	0.71
Plateletcrit (%)	0.36 ± 0.23	0.66 ± 0.60	0.01–9.99	0.32
Platelet/large cell ratio (%)	21.54 ± 8.26	28.88 ± 13.49	0.1–99.9	0.32

^1^ Different lowercase letters (a,b) indicate significant differences between the CON and DNC groups (*p* ≤ 0.05) as determined via the Student–Newman–Keuls test. The values are expressed as means ± SD, *n* = 6. CON, control diet, dietary nutritional combinations not added, and DNC, addition of dietary nutritional combinations.

**Table 4 animals-14-01949-t004:** Effect of added DNC on blood biochemical indices in kittens.

Items	CON	DNC	Normal Reference Range	*p*-Value
Albumin (g/L)	35.70 ± 3.36	31.97 ± 2.58	22.0–43.0	0.06
Total Protein (g/L)	64.45 ± 8.36	55.97 ± 7.41	53.0–82.0	0.09
Globulin (g/L)	28.85 ± 5.14	24.03 ± 3.65	23.0–48.0	0.09
Albumin/Globulin Ratio	1.24 ± 0.21	1.34 ± 0.34	-	0.96
Aspartate Aminotransferase (U/L)	22.14 ± 3.86 ^a^	12.51 ± 5.20 ^b^	0.0–48.0	<0.01
Alanine Aminotransferase (U/L)	65.17 ± 12.68	58.33 ± 11.59	5–115	0.36
Amylase (U/L)	449.50 ± 52.50	478.46 ± 36.45	450–1500	0.30
Creatine Kinase (CK) (U/L)	502.42 ± 95.31 ^a^	373.31 ± 65.22 ^b^	0–467	<0.05
Creatinine (umol/L)	45.90 ± 7.35	50.03 ± 6.74	25.0–141.0	0.33
Blood Urea Nitrogen (mmol/L)	10.28 ± 4.87	12.12 ± 3.85	4.00–12.80	0.48
BUN/Creatinine Ratio	308.81 ± 66.37	242.97 ± 42.34	-	0.07
Glucose (mmol/L)	6.35 ± 0.56	5.83 ± 0.21	4.28–8.50	0.06
Triglycerides (mmol/L)	1.54 ± 0.42	1.98 ± 0.54	0.00–2.20	0.14
Calcium (mmol/L)	2.80 ± 0.44	2.66 ± 0.36	1.98–2.83	0.57
Inorganic Phosphorus (umol/L)	3.21 ± 0.03	3.23 ± 0.09	1.45–3.35	0.67
Total Bilirubin (mmol/L)	1.70 ± 0.24	1.30 ± 0.47	0.0–15.0	0.09

^1^ Different lowercase letters (a,b) indicate significant differences between the CON and DNC groups (*p* ≤ 0.05) as determined via the Student–Newman–Keuls test. The values are expressed as means ± SD, *n* = 6. CON, control diet, dietary nutritional combinations not added, and DNC, addition of dietary nutritional combinations.

**Table 5 animals-14-01949-t005:** α-Diversity indices of the fecal microbiota in kittens fed with DNC-supplemented feed.

Items	CON 0	CON 28	DNC 0	DNC 28	*p*-Value
Ace	100.50 ± 35.18 ^a^	106.80 ± 8.95 ^a^	109.00 ± 8.10 ^a^	56.67 ± 10.55 ^b^	<0.01
Chao1	101.60 ± 35.35 ^a^	103.90 ± 9.63 ^a^	110.8 ± 10.97 ^a^	54.27 ± 8.12 ^b^	<0.01
Coverage	1.00 ± 0.00 ^a^	1.00 ± 0.00 ^a^	1.00 ± 0.00 ^a^	1.00 ± 0.00 ^a^	0.77
Shannon	2.12 ± 0.80 ^a^	2.19 ± 0.28 ^a^	2.16 ± 0.37 ^a^	1.67 ± 0.22 ^b^	0.06
Simpson	0.26 ± 0.23 ^a^	0.22 ± 0.11 ^a^	0.23 ± 0.09 ^a^	0.28 ± 0.07 ^a^	0.71
Sobs	93.60 ± 33.02 ^a^	94.60 ± 10.60 ^a^	96.20 ± 7.59 ^a^	50.80 ± 8.67 ^b^	<0.01

^1^ Different lowercase letters (a,b) indicate significant differences among all groups (*p* ≤ 0.05) as determined via the Student–Newman–Keuls test. The values are expressed as means ± SD, *n* = 6. CON0, control diet, dietary nutritional combinations not added at day 0; DNC0, addition of dietary nutritional combinations at day 0; CON28, control diet, dietary nutritional combinations not added at day 28; and DNC28, addition of dietary nutritional combinations at day 28.

## Data Availability

The data generated from the study is clearly presented and discussed in the manuscript.

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
