# Peer review of "Effects of Lactoferrin and Lactobacillus Supplementation on Immune Function, Oxidative Stress, and Gut Microbiota in Kittens"

_animals, 2024, doi:10.3390/ani14131949_

Round 1
Reviewer 1 Report
Comments and Suggestions for Authors
Dear Authors, the topic of this study is extremely relevant, and I thank you for the idea of ​​this research. Neonatology/small animal pediatrics will greatly benefit from studies like this.
This research could make a significant contribution to the literature and routine practice in kittens. However, there are some topics to review. My main questions and suggestions are described below.
ABSTRACT
The study has promising results, please include more important results information in the abstract.
INTRODUCTION
Lines 50-85 - Please make it clear to the reader which species the studies mentioned are talking about, the majority are from humans and when this is not explicit it can lead to confusion, presuming that they are studies on animals/kittens.
MATERIALS AND METHODS
Lines 139-141- ‘blood specimens were collected into serum separator tubes containing ethylenediaminetetraacetic acid (EDTA) anticoagulant, subsequently allowed to coagulate at 28°C for 30 minutes, and thereafter subjected to centrifugation’
If blood serum was obtained, the tube used does not contain anticoagulant (EDTA), please correct this information.
Were no blood samples taken on day 0? To compare the parameters with the day 28?
RESULTS
Although few animals were used for the study, were differences observed in the parameters evaluated between 2-month-old and 3-month-old animals? This is important, as there are hematological, biochemical and immunoglobulin concentration differences between these ages. For example, at two months of age the kitten's immune system is more immature than at 3 months, which can impact immunoglobulin production responses. This should also be mentioned in the discussion.
Lines 322-332 – These parameters were not explained in the material and methods, please include them.
DISCUSSION
Lines 461-471 – Please, explain to the reader the mechanisms by which lactoferrin and probiotics increase the production of immunoglobulins. For example, it is known that the intestinal microbiota is essential for the development of gut-associated lymphoid tissue (GALT), such as Peyer's Patches. GALT is the main site of proliferation of B lymphocytes (which can differentiate into plasma cells, which secrete immunoglobulins).
Author Response
Thank you very much for taking the time to review our manuscript titled Lactoferrin and Lactobacillus Supplementation on Immune Function, Oxidative Stress, and Gut Microbiota in Kittens (animals 3070105). We greatly appreciate your detailed and constructive comments, which have significantly helped us to improve the quality of our work. The manuscript has been revised in accordance with your suggestions, and the updated version can be found in the attached document.

Reviewer 2 Report
Comments and Suggestions for Authors
This interesting study examines the effects of dietary supplements containing lactoferrin and Lactobacillus supplements immune function, gut health, and microbiota composition in kittens. The results show significant improvements: a 14.9% increase in IgA levels, a 14.2% increase in IgG levels, a 28.7% increase in catalase activity, and improved gastrointestinal health with more beneficial bacteria. These results highlight lactoferrin and probiotics as effective dietary supplements to boost kitten health, reduce antibiotic dependency, and improve overall vitality during critical growth phases. The reviewer thanks the authors for their work; however, some important comments need to be considered.
General comments
Firstly, all the information on lactobacilli has been extrapolated from species such as humans, and mice, which are omnivores (with important fermentation activity in their gut, particularly in the colon), whereas cat are known strict carnivores. The considerable difference between these species is important, as some species of lactobacilli are very important in the colon of some hosts, whereas in carnivores with a shorter GI tract and faster GI transit, and more enzymatic digestion rather than fermentation, little effect has been reported.
The authors state that the microbiota improves, but fundamental markers such as diversity, richness and Blautia decrease considerably after the use of this intervention on day 28. Blautia is in fact considered a core bacterial genus in the microbiota and has been included in faecal dysbiosis indices as an indicator of GI health. What is actually observed is an increase in Firmicutes (Bacteroidota), due to the massive intake of Lactobacillus and consequently its detection in the faeces. But what would happen if this probiotic were stopped? Would these marked changes continue? In the reviewer's opinion, observing these changes is not indicative of an improvement in the microbiota. On the other hand, the authors should have made the measurements of all the variables (and not only microbiota) at time 0, and subsequently at time 1 (day 28). We do not know if there were basal differences in the groups, as their criterion for classifying them was weight. Therefore, we do not know if they could have a different baseline level and that when this diet supplemented was introduced, without any changes, differences were observed at 28 days, which were not a direct effect of the Lactobacillus plus ferritin. What about the weight when the study ends? Still no difference between the groups?
In addition, the authors have used a correlation test assuming a normal distribution of the percentages (relative abundance) of bacterial genera. However, given the low sample size and the high diversity present in the microbiota, it is almost certain that the data do not follow a normal distribution. The authors do not present a test for normality of the correlated variables, and therefore, it would be correct to assume that they do not follow a normal distribution and the correct test would be Spearman's correlation, rather than Pearson's correlation.
Finally, some limitations of the study that have not been mentioned in the Discussion section. These include the low sample size (considering the large individual variability in the microbiota profile), as well as the combined use of Lactobacillus and lactoferrin, which makes it impossible to assess which is responsible for the observed changes.
Specific comments
Simple summary is lacking
Abstract:
· Line 18: which species, and more importantly, which strains as some probiotic functions are strain specific. This information is essential.
· Line 22: which beneficial bacterial populations?
Introduction
· Lines 68-70: in which animals/human species has it been shown to significantly improve gut performance? Reference studies in cats, as significant differences have been found between core microbiota of humans and cats, and it is well known that cats are strict carnivores, and different species can assume the same role (for example: obtain SCFA from protein, instead of carbohydrates).
· Line 81: in adults_[23] (the space).
· Line 83: in stressed adults of which species? Reference 24 is incomplete.
· Line 90: Please specify which species and specific strains of Lactobacillus, as it is well known that most of the functions of probiotics are strain specific. Thus, even if they belong to the same genera the great variability of species could lead to a bias. In line 110, the authors stated that they use 1x1010 CFU/Kg L. plantarum. Which strain?
Material and Methods
· Line 105: 4 males and 6 females, making a total of 10 animals. The authors stated that there were 12 cats. 6 males and 6 females? Please, check this information.
· Line 106 and 110: “839 ± 75.3 g” and “no significant difference in the mean body weight of the kittens in each group” are considered by the reviewer to be results.
· Line 114: I suggest replacing with “microbiota” (instead of “flora”). Flora is a botanical term that was used to classify everything that was not part of the animal kingdom. Therefore, although the term is widely used, it is not correct.
· Line 118: “ad libitum” in italics to indicate that it is a foreign expression (from Latin).
· Line 144: “blood biochemistry analyzer” (which one). Please, specify.
· Line 155: Please remove the numbering at the beginning of the paragraphs that goes from 1 to 6.
· Line 178-179: ImageJ software (which version of the software, city and country of the developer) or reference “Schindelin, J.; Arganda-Carreras, I.; Frise, E.; Kaynig, V.; Longair, M.; Pietzsch, T.; Preibisch, S.; Rueden, C.; Saalfeld, S.; Schmid, B. Fiji: An open-source platform for biological-image analysis. Nat. Methods 2012, 9, 676–682.” On the other hand, which features were measured?
· Line 182: Again, I suggest microbiota rather than flora. 16S rARN gene.
· Line 190: Please specify the kit and company or the method for the DNA extraction.
· Line 191: How? Nanodrop? Using A260/280; A260/A230?
· Line 218: What about the correlations? The authors used a Pearson correlation but given the small sample size and the large variability in the relative abundance of different genera, the authors should have used a Spearman correlation test, as the data probably did not follow a normal distribution and a non-parametric approach is more powerful.
Results
· Table 3. If all the paired variables evaluated did not present differences, it may be better to indicate only those that are different with a lower lowercase letter. On the other hand, it is important to indicate whether the values obtained for total white blood cells are in the normal range, even if they are statistically significantly different, because they can be different but still in the range. I encourage to the authors to add a column with the normal range values to make it easier to interpret the results.
· Lines 257-264: Again, it is important to consider whether these significant statistical differences imply a biological value or whether they are still within the normal range.
· Figure 2: Please improve the resolution of the image.
· Figure 6: Again, I suggest improving the resolution of the pictures, as information in the legend cannot be visualized.
· Line 353: More than one index indicates a lower diversity of fecal microbiota. How does the author explain this as a better result?
· Line 396: Enterococcus in italics.
· Line 409-417: Please, when expressing correlations, the authors should also include the P-value, not just “r”.
· Line 415: Collinsella in Italics
· Figure 7: Again, Pearson is not the correct statistical test as the variables do not follow the normality. If the authors confirm that all parameters follow normal distribution, what test did they use to prove this? This information is missing in M&M, in the statistical analysis.
Discussion
· Line 539-540: “marked reduction in […] indicates a shift towards a healthier gut microbiota composition” based on what evidence? Is more important the function rather than the genera involved. In addition, Blautia has been included as a GI health biomarker in a well-known dysbiosis index in cats.
Data availability Statement: the sequences of the 16SrARN should be available in a public repository.
References: Please check the references according to the journal style.
· Line 655: Reference 24 is incomplete.
Author Response

(The authors gave the same response as above.)

Round 2
Reviewer 2 Report
Comments and Suggestions for Authors
The reviewer would like to congratulate the authors on their work done. Most of the suggestions have been addressed. However, some minor details are still pending. Please make sure that all the changes listed in the reviewer's response letter have been addressed in a similar way in the manuscript:
· Line 35: Lactobacillus (in italics).
· Line 40: L. plantarum also in italics
· Line 74: animals. [13]. Remove the period between “s” and “[“.
· Line 107: As the authors refer to mice, the correct term is “clinical signs” rather than “symptoms”.
· Line 118: The authors have specified the species of Lactobacillus (L. plantarum). However, they have still not specified the strain. As they have already shown, when using a probiotic (e.g. Fortiflora, they refer to Enterococcus faecium SF68, where SF68 refers to the strain name. However, there is no information on the strain is in the manuscript. This is an important limitation.
· Line 215: 16S rARN gene.
· Line 492: day 0, instead of “Day 0”
· Line 518, 522-524: production_[39]; production_[40], response_[41]. (The spaces).
· Line 609: Lactobacillus (in italics). I also suggest adding a reference showing that Blautia is a core genus.